# Value Distribution and Arbitrary-Order Derivatives of Meromorphic Solutions of Complex Linear Differential Equations in the Unit Disc

## Hai-Ying Chen and Xiu-Min Zheng *

Institute of Mathematics and Information Science, Jiangxi Normal University, Nanchang 330022, Jiangxi, China; chenhaiying182@126.com
* Correspondence: zhengxiumin2008@sina.com

**Abstract:** In this paper, we investigate the value distribution of meromorphic solutions and their arbitrary-order derivatives of the complex linear differential equation $f'' + A(z)f' + B(z)f = F(z)$ in $\Delta$ with analytic or meromorphic coefficients of finite iterated $p$-order, and obtain some results on the estimates of the iterated exponent of convergence of meromorphic solutions and their arbitrary-order derivatives taking small function values.

**Keywords:** complex linear differential equation; unit disc; meromorphic solution; arbitrary-order derivative; small function

**MSC:** 30D35; 34M10

## 1. Introduction and Main Results

Throughout this paper, we use the standard notations of the classic Nevanlinna theory (see, e.g., [1,2]), such as $m(r, f), n(r, f), N(r, f), T(r, f)$, and $M(r, f)$.

In the following, we denote the whole complex plane as $\mathbf{C}$ and denote the unit disc as $\Delta = \{z \in \mathbf{C} : |z| < 1\}$; we denote $\mathbf{N}_+ = \{1, 2, \cdots\}$, and assume that $p, k \in \mathbf{N}_+$ unless otherwise specified. We also define inductively [3], for $r \in [0, +\infty)$, $\exp_1 r = \exp r$, $\exp_{p+1} r = \exp(\exp_p r)$; and, for sufficiently large $r \in (0, +\infty)$, $\log_1 r = \log r$, $\log_{p+1} r = \log(\log_p r)$; additionally, we denote $\exp_0 r = r = \log_0 r$, $\exp_{-1} r = \log_1 r = \log r$, $\log_{-1} r = \exp_1 r = \exp r$.

Firstly, we introduce some definitions on the growth and the value distribution of fast-growing meromorphic functions in $\Delta$ (see, e.g., [4–9]).

**Definition 1** ([6]). *Let $f(z)$ be a meromorphic function in $\Delta$. Then, we define the iterated $p$-order of $f(z)$ as*

$$\sigma_p(f) = \varlimsup_{r \to 1^-} \frac{\log_p^+ T(r, f)}{\log \frac{1}{1-r}}.$$

*If $f(z)$ is analytic in $\Delta$, we also define*

$$\sigma_{M,p}(f) = \varlimsup_{r \to 1^-} \frac{\log_{p+1}^+ M(r, f)}{\log \frac{1}{1-r}}.$$

**Remark 1.** *From Tsuji [10] and Laine [2], respectively, we can see that if $f(z)$ is analytic in $\Delta$, then we have*

$$\sigma_1(f) \leq \sigma_{M,1}(f) \leq \sigma_1(f) + 1$$

*and*

$$\sigma_p(f) = \sigma_{M,p}(f) \quad (p \in \mathbf{N}_+ \setminus \{1\}).$$

**Definition 2** ([6,7]). *Let $f(z)$ be a meromorphic function in $\Delta$. Then, we define the growth index of the iterated order of $f(z)$ as*

$$i(f) = \begin{cases} 0, & \text{if } f(z) \text{ is non-admissible;} \\ \min\{p : \sigma_p(f) < \infty\}, & \text{if } f(z) \text{ is admissible and } \sigma_p(f) < \infty \text{ for some } p; \\ \infty, & \text{if } \sigma_p(f) = \infty \text{ for all } p. \end{cases}$$

*If $f(z)$ is analytic in $\Delta$, we also define*

$$i_M(f) = \begin{cases} 0, & \text{if } f(z) \text{ is non-admissible;} \\ \min\{p : \sigma_{M,p}(f) < \infty\}, & \text{if } f(z) \text{ is admissible and } \sigma_{M,p}(f) < \infty \text{ for some } p; \\ \infty, & \text{if } \sigma_{M,p}(f) = \infty \text{ for all } p. \end{cases}$$

**Definition 3** ([4,5]). *Let $f(z)$ be a meromorphic function in $\Delta$. Then, we define the iterated p-exponent of convergence of the sequence of zeros of $f(z)$ and the iterated p-exponent of convergence of the sequence of distinct zeros of $f(z)$, respectively, as*

$$\lambda_p(f) = \varlimsup_{r \to 1^-} \frac{\log_p^+ N(r, \frac{1}{f})}{\log \frac{1}{1-r}}$$

*and*

$$\overline{\lambda}_p(f) = \varlimsup_{r \to 1^-} \frac{\log_p^+ \overline{N}(r, \frac{1}{f})}{\log \frac{1}{1-r}}.$$

**Definition 4** ([9]). *Let $f(z)$ be a meromorphic function in $\Delta$ with the iterated p-order $\sigma_p(f)(0 < \sigma_p(f) < \infty)$. Then, we define the iterated p-type of $f(z)$ as*

$$\tau_p(f) = \varlimsup_{r \to 1^-} \frac{\log_{p-1}^+ T(r, f)}{(\frac{1}{1-r})^{\sigma_p(f)}}.$$

*If $f(z)$ is analytic in $\Delta$, we also define*

$$\tau_{M,p}(f) = \varlimsup_{r \to 1^-} \frac{\log_p^+ M(r, f)}{(\frac{1}{1-r})^{\sigma_{M,p}(f)}}.$$

**Definition 5** ([1]). *Let $f(z)$ be a meromorphic function in $\Delta$. Then, for $a \in \overline{\mathbf{C}} = \mathbf{C} \cup \{\infty\}$, we define the deficiency of the value a with respect to $f(z)$ as*

$$\delta(a, f) = \varliminf_{r \to 1^-} \frac{m(r, \frac{1}{f-a})}{T(r, f)}.$$

Next, we introduce some background relative to our main results. It is well-known that Bank and Laine started the original complex oscillation theory of solutions of linear differential equations in **C** in 1982 (see [2]). Following that, many scholars in the field of complex analysis have investigated the growth and the value distribution of meromorphic solutions of complex linear differential equations as the theory of complex linear differential equations in **C** has matured (see, e.g., [2,3,11,12]). Naturally, the question of whether we can get the corresponding results on complex linear differential equations in $\Delta$ has arisen. This question is interesting and meaningful. On the one hand, complex linear differential equations in $\Delta$ have many similar properties to those in **C**. On the other hand, it is much more difficult

to study complex linear differential equations in $\Delta$ than in $\mathbf{C}$, due to the lack of corresponding effective tools. Some results on this topic can be seen in, for example, [4–9,13–20].

In particular, Latreuch and Belaïdi [18] investigated the distribution of zeros of meromorphic solutions and their arbitrary-order derivatives for a second-order non-homogeneous complex linear differential equation

$$f'' + A(z)f' + B(z)f = F(z) \tag{1}$$

in $\Delta$ with meromorphic coefficients of finite iterated $p$-order, and got the following result:

**Theorem 1** ([18]). *Let $A(z), B(z)(\not\equiv 0)$, and $F(z)(\not\equiv 0)$ be meromorphic functions in $\Delta$ with finite iterated $p$-order, such that $B_j(z) \not\equiv 0$ and $F_j(z) \not\equiv 0$ $(j = 1, 2, \cdots)$, where $B_j(z)$ and $F_j(z)(j = 1, 2, \cdots)$ are defined as follows*

$$A_0(z) = A(z), \quad A_j(z) = A_{j-1}(z) - \frac{B'_{j-1}(z)}{B_{j-1}(z)} \quad (j = 1, 2, \cdots), \tag{2}$$

$$B_0(z) = B(z), \quad B_j(z) = A'_{j-1}(z) - A_{j-1}(z)\frac{B'_{j-1}(z)}{B_{j-1}(z)} + B_{j-1}(z) \quad (j = 1, 2, \cdots), \tag{3}$$

$$F_0(z) = F(z), \quad F_j(z) = F'_{j-1}(z) - F_{j-1}(z)\frac{B'_{j-1}(z)}{B_{j-1}(z)} \quad (j = 1, 2, \cdots). \tag{4}$$

*(a) If $f(z)$ is a meromorphic solution of (1) in $\Delta$ with $\sigma_p(f) = \infty$ and $\sigma_{p+1}(f) = \sigma$, then $f(z)$ satisfies*

$$\overline{\lambda}_p(f^{(j)}) = \lambda_p(f^{(j)}) = \sigma_p(f) = \infty \quad (j = 0, 1, \cdots)$$

*and*

$$\overline{\lambda}_{p+1}(f^{(j)}) = \lambda_{p+1}(f^{(j)}) = \sigma_{p+1}(f) = \sigma \quad (j = 0, 1, \cdots).$$

*(b) If $f(z)$ is a meromorphic solution of (1) in $\Delta$ with*

$$\sigma_p(f) > \max\{\sigma_p(A), \sigma_p(B), \sigma_p(F)\},$$

*then $f(z)$ satisfies*

$$\overline{\lambda}_p(f^{(j)}) = \lambda_p(f^{(j)}) = \sigma_p(f) \quad (j = 0, 1, \cdots).$$

They also noted that some special conditions on the coefficients in (1) can guarantee that the assumptions $B_j(z) \not\equiv 0$ and $F_j(z) \not\equiv 0$ $(j = 1, 2, \cdots)$ in Theorem 1 hold, which makes Theorem 1 more concrete. More details can be seen in Theorems 2 and 3.

**Theorem 2** ([18]). *Let $A(z)$, $B(z)(\not\equiv 0)$, and $F(z)(\not\equiv 0)$ be analytic functions in $\Delta$ with finite iterated p-order, such that $\sigma_p(B) > \max\{\sigma_p(A), \sigma_p(F)\}$. Then, all non-trivial solutions of (1) in $\Delta$ satisfy*

$$\sigma_p(B) \leq \overline{\lambda}_{p+1}(f^{(j)}) = \lambda_{p+1}(f^{(j)}) = \sigma_{p+1}(f) \leq \sigma_{M,p}(B) \quad (j = 0, 1, \cdots),$$

*with at most one possible exceptional solution, $f_0(z)$, such that*

$$\sigma_{p+1}(f_0) < \sigma_p(B).$$

**Theorem 3** ([18])**.** *Let $A(z)$, $B(z)(\not\equiv 0)$, and $F(z)(\not\equiv 0)$ be meromorphic functions in $\Delta$ with finite iterated p-order, such that $\sigma_p(B) > \max\{\sigma_p(A), \sigma_p(F)\}$. If $f(z)$ is a meromorphic solution of (1) in $\Delta$ with $\sigma_p(f) = \infty$ and $\sigma_{p+1}(f) = \sigma$, then $f(z)$ satisfies*

$$\overline{\lambda}_p(f^{(j)}) = \lambda_p(f^{(j)}) = \sigma_p(f) = \infty \quad (j = 0, 1, \cdots)$$

*and*

$$\overline{\lambda}_{p+1}(f^{(j)}) = \lambda_{p+1}(f^{(j)}) = \sigma_{p+1}(f) = \sigma \quad (j = 0, 1, \cdots),$$

*where*

$$\sigma_p(f) = \overline{\lim_{r \to 1^-}} \frac{\log_p m(r, f)}{\log \frac{1}{1-r}}.$$

Later, Gong and Xiao [17] generalized Theorems 1–3, and obtained the following results which consider the distribution of meromorphic solutions and their arbitrary-order derivatives taking small function values instead of taking zeros.

**Theorem 4** ([17])**.** *Let $A(z)$, $B(z)(\not\equiv 0)$, $F(z)(\not\equiv 0)$, and $\varphi(z)$ be meromorphic functions in $\Delta$ with finite iterated p-order, such that $B_j(z) \not\equiv 0$ and $D_j(z) \not\equiv 0$ $(j = 0, 1, \cdots)$, where $B_j(z)$ and $D_j(z)(j = 0, 1, \cdots)$ are defined by (2)–(4) and the following*

$$D_j(z) = F_j(z) - (\varphi''(z) + A_j(z)\varphi'(z) + B_j(z)\varphi(z)) \quad (j = 0, 1, \cdots). \tag{5}$$

*(a) If $f(z)$ is a meromorphic solution of (1) in $\Delta$ with $\sigma_p(f) = \infty$ and $\sigma_{p+1}(f) = \sigma$, then $f(z)$ satisfies*

$$\overline{\lambda}_p(f^{(j)} - \varphi) = \lambda_p(f^{(j)} - \varphi) = \sigma_p(f) = \infty \quad (j = 0, 1, \cdots)$$

*and*

$$\overline{\lambda}_{p+1}(f^{(j)} - \varphi) = \lambda_{p+1}(f^{(j)} - \varphi) = \sigma_{p+1}(f) = \sigma \quad (j = 0, 1, \cdots).$$

*(b) If $f(z)$ is a meromorphic solution of (1) in $\Delta$ with*

$$\max\{\sigma_p(A), \sigma_p(B), \sigma_p(F), \sigma_p(\varphi)\} < \sigma_p(f) < \infty,$$

*then $f(z)$ satisfies*

$$\overline{\lambda}_p(f^{(j)} - \varphi) = \lambda_p(f^{(j)} - \varphi) = \sigma_p(f) \quad (j = 0, 1, \cdots).$$

Similar to Theorems 2 and 3, they also obtained more concrete results corresponding Theorem 4, as follows in Theorems 5 and 6.

**Theorem 5** ([17])**.** *Let $A(z)$, $B(z)(\not\equiv 0)$, $F(z)(\not\equiv 0)$, and $\varphi(z)$ be analytic functions in $\Delta$ with finite iterated p-order, such that $\sigma_p(B) > \max\{\sigma_p(A), \sigma_p(F), \sigma_p(\varphi)\}$, $\sigma_{M,p}(A) \le \sigma_{M,p}(B)$, and $\varphi(z)$ is not a solution of (1). Then, all non-trivial solutions of (1) in $\Delta$ satisfy*

$$\sigma_p(B) \le \overline{\lambda}_{p+1}(f^{(j)} - \varphi) = \lambda_{p+1}(f^{(j)} - \varphi) = \sigma_{p+1}(f) \le \sigma_{M,p}(B) \quad (j = 0, 1, \cdots),$$

*with at most one possible exceptional solution, $f_0(z)$, such that*

$$\sigma_{p+1}(f_0) < \sigma_p(B).$$

**Theorem 6** ([17]). *Let $A(z)$, $B(z)(\not\equiv 0)$, $F(z)(\not\equiv 0)$, and $\varphi(z)$ be meromorphic functions in $\Delta$ with finite iterated p-order, such that $\sigma_p(B) > \max\{\sigma_p(A), \sigma_p(F), \sigma_p(\varphi)\}$, $\delta(\infty, B) > 0$, and $\varphi(z)$ is not a solution of (1). If $f(z)$ is a meromorphic solution of (1) in $\Delta$ with $\sigma_p(f) = \infty$ and $\sigma_{p+1}(f) = \sigma$, then $f(z)$ satisfies*

$$\overline{\lambda}_p(f^{(j)} - \varphi) = \lambda_p(f^{(j)} - \varphi) = \sigma_p(f) = \infty \quad (j = 0, 1, \cdots)$$

*and*

$$\overline{\lambda}_{p+1}(f^{(j)} - \varphi) = \lambda_{p+1}(f^{(j)} - \varphi) = \sigma_{p+1}(f) = \sigma \quad (j = 0, 1, \cdots).$$

Inspired by Theorems 1–6, we proceed further in this direction. Note that there exists a dominant coefficient whose iterated *p*-order is strictly larger than those of the other coefficients in Theorems 2, 3, 5, and 6. A question arises naturally: What can we say if there exists more than one coefficient having the maximal iterated *p*-order? In the following, we introduce a condition on the iterated *p*-type to deal with coefficients having the maximal iterated *p*-order to obtain Theorems 7 and 8.

**Theorem 7.** *Let $p \in \mathbf{N}_+ \backslash \{1\}$, $A(z)$, $B(z)(\not\equiv 0)$, $F(z)(\not\equiv 0)$, and $\varphi(z)(\not\equiv 0)$ be analytic functions in $\Delta$ with finite iterated p-order, such that $\max\{\sigma_p(A), \sigma_p(F), \sigma_p(\varphi)\} \leq \sigma_p(B)(0 < \sigma_p(B) < \infty)$, $\max\{\tau_p(S) : \sigma_p(S) = \sigma_p(B), S \in \{A, F, \varphi\}\} < \tau_p(B) < \infty$, $\sigma_{M,p}(A) \leq \sigma_{M,p}(B)$, and $\varphi(z)$ is not a solution of (1). Then, all non-trivial solutions $f(z)$ of (1) in $\Delta$ satisfy*

$$\sigma_p(B) = \overline{\lambda}_{p+1}(f^{(j)} - \varphi) = \lambda_{p+1}(f^{(j)} - \varphi) = \sigma_{p+1}(f) = \sigma_{M,p+1}(f) = \sigma_{M,p}(B) \quad (j = 0, 1, \cdots),$$

*with at most one possible exceptional solution, $f_0(z)$, satisfying*

$$\sigma_{M,p+1}(f_0) = \sigma_{p+1}(f_0) < \sigma_p(B).$$

**Remark 2.** *The partial result of Theorem 7 for the case $p = 1$ will be shared in Lemma 7.*

**Theorem 8.** *Let $p \in \mathbf{N}_+ \backslash \{1\}$, $A(z), B(z)(\not\equiv 0)$, $F(z)(\not\equiv 0)$, and $\varphi(z)$ be meromorphic functions in $\Delta$ with finite iterated p-order, such that $\max\{\sigma_p(A), \sigma_p(F), \sigma_p(\varphi)\} \leq \sigma_p(B)(0 < \sigma_p(B) < \infty)$, $\max\{\tau_p(S) : \sigma_p(S) = \sigma_p(B), S \in \{A, F, \varphi\}\} < \tau_p(B) < \infty$, $\delta(\infty, B) > 0$, and $\varphi(z)$ is not a solution of (1). If $f(z)$ is a meromorphic solution of (1) in $\Delta$ with $\sigma_p(f) = \infty$ and $\sigma_{p+1}(f) < \infty$, then $f(z)$ satisfies*

$$\overline{\lambda}_p(f^{(j)} - \varphi) = \lambda_p(f^{(j)} - \varphi) = \sigma_p(f) = \infty \quad (j = 0, 1, \cdots)$$

*and*

$$\overline{\lambda}_{p+1}(f^{(j)} - \varphi) = \lambda_{p+1}(f^{(j)} - \varphi) = \sigma_{p+1}(f) \quad (j = 0, 1, \cdots).$$

**Remark 3.** *The partial results of Theorem 8 for the case $p = 1$ can be seen in Lemmas 2 and 3.*

## 2. Lemmas for Proofs of Main Results

**Lemma 1** ([14]). *Let $f(z)$ be a meromorphic function in $\Delta$ with $i(f) = p$ and $\sigma_p(f) < \infty$. Then, for any $\varepsilon(> 0)$, there exists a subset $E \subset [0, 1)$ with $\int_E \frac{dr}{1-r} < \infty$ such that, for all $r \notin E, r \to 1^-$, we have*

$$m\left(r, \frac{f^{(k)}}{f}\right) = O\left(\exp_{p-2}\left\{\left(\frac{1}{1-r}\right)^{\sigma_p(f)+\varepsilon}\right\}\right).$$

**Lemma 2** ([18]). *Let $A_0(z), A_1(z), \cdots, A_{k-1}(z)$, and $F(z)(\not\equiv 0)$ be meromorphic functions in $\Delta$ with finite iterated p-order. If $f(z)$ is a meromorphic solution of complex linear differential equation*

$$f^{(k)} + A_{k-1}(z)f^{(k-1)} + \cdots + A_0(z)f = F(z) \tag{6}$$

*in* $\mathbf{\Delta}$ *with* $\sigma_p(f) = \infty$ *and* $\sigma_{p+1}(f) < \infty$, *then* $f(z)$ *satisfies*

$$\overline{\lambda}_p(f) = \lambda_p(f) = \sigma_p(f) = \infty$$

*and*

$$\overline{\lambda}_{p+1}(f) = \lambda_{p+1}(f) = \sigma_{p+1}(f).$$

**Lemma 3** ([17]). *Let* $A_0(z), A_1(z), \cdots, A_{k-1}(z), F(z)(\not\equiv 0)$ *and* $\varphi(z)$ *be meromorphic functions in* $\mathbf{\Delta}$ *with finite iterated p-order, such that*

$$F(z) - \varphi^{(k)}(z) - A_{k-1}(z)\varphi^{(k-1)}(z) - \cdots - A_1(z)\varphi'(z) - A_0(z)\varphi(z) \not\equiv 0.$$

*If* $f(z)$ *is a meromorphic solution of (6) in* $\mathbf{\Delta}$ *with* $\sigma_p(f) = \infty$ *and* $\sigma_{p+1}(f) < \infty$, *then* $f(z)$ *satisfies*

$$\overline{\lambda}_p(f - \varphi) = \lambda_p(f - \varphi) = \sigma_p(f) = \infty$$

*and*

$$\overline{\lambda}_{p+1}(f - \varphi) = \lambda_{p+1}(f - \varphi) = \sigma_{p+1}(f).$$

**Lemma 4** ([11]). *Let* $A_0(z), A_1(z), \cdots, A_{k-1}(z)$ *be analytic functions in* $D_R(= \{z \in \mathbf{C} : |z| < R\})$, *where* $0 < R \le \infty$, *and* $f(z)$ *be a solution of complex linear differential equation*

$$f^{(k)} + A_{k-1}(z)f^{(k-1)} + \cdots + A_0(z)f = 0$$

*in* $D_R$. *Then, for all* $0 \le r < R$,

$$m_p(r, f)^p \le C\left(\sum_{j=0}^{k-1} \int_0^{2\pi} \int_0^r |A_j(se^{i\theta})|^{\frac{p}{k-j}} ds d\theta + 1\right),$$

*where* $C = C(k) > 0$ *is a constant depending on p and on the initial values of* $f(z)$ *at the point* $z_\theta$, *where* $A_j(z_\theta) \ne 0$ *for some* $j = 0, 1 \cdots, k - 1$.

**Lemma 5** ([19]). *Let* $f(z)$ *be an analytic function in* $\mathbf{\Delta}$ *with finite iterated order, such that* $\sigma_{M,p}(f) = \sigma > 0$, $0 < \tau_{M,p}(f) = \tau < \infty$. *Then, for any* $\mu(< \tau)$, *there exists a subset* $H \subset [0, 1)$ *with* $\int_H \frac{dr}{1-r} = \infty$ *such that for all* $r \in H$, *we have*

$$\log_p M(r, f) > \mu\left(\frac{1}{1-r}\right)^\sigma.$$

**Remark 4.** *If the definitions of* $\sigma_{M,p}(f)$ *and* $\tau_{M,p}(f)$ *in Lemma 5 are replaced by* $\sigma_p(f)$ *and* $\tau_p(f)$, *respectively, it is obvious that, for any* $\varepsilon(0 < \varepsilon < \tau_p(f))$, *there exists a subset* $H \subset [0, 1)$ *with* $\int_H \frac{dr}{1-r} = \infty$ *such that, for all* $r \in H$, *we have*

$$\log_{p-1} T(r, f) > (\tau_p(f) - \varepsilon)\left(\frac{1}{1-r}\right)^{\sigma(f)}.$$

**Lemma 6.** *Let* $A(z)$ *and* $B(z)$ *be analytic functions in* $\mathbf{\Delta}$ *such that* $\sigma_p(A) \le \sigma_p(B)(0 < \sigma_p(B) < \infty)$, $\tau_p(A) < \tau_p(B)$ *if* $\sigma_p(A) = \sigma_p(B)$, $\sigma_{M,p}(A) \le \sigma_{M,p}(B)$. *If* $f(z)(\not\equiv 0)$ *is a solution of*

$$f'' + A(z)f' + B(z)f = 0 \tag{7}$$

*in* $\mathbf{\Delta}$, *then* $f(z)$ *satisfies*

$$\sigma_p(B) \le \sigma_{p+1}(f) = \sigma_{M,p+1}(f) \le \sigma_{M,p}(B).$$

*Further, if $p \in \mathbf{N}_+\backslash\{1\}$, then $f(z)$ satisfies*

$$\sigma_p(B) = \sigma_{p+1}(f) = \sigma_{M,p+1}(f) = \sigma_{M,p}(B).$$

**Proof.** We divide this proof into two parts.

Firstly, we prove $\sigma_{p+1}(f) \leq \sigma_{M,p}(B)$. By Lemma 4, for $r \in (0,1)$, we have

$$
\begin{aligned}
T(r,f) = m(r,f) \quad &\leq C[\int_0^{2\pi} \int_0^r (|A(se^{i\theta})| + |B(se^{i\theta})|^{\frac{1}{2}}) ds d\theta + 1] \\
&\leq 2\pi C[M(r,A) + M(r,B) + 1],
\end{aligned}
\tag{8}
$$

where $C = C(k) > 0$ is a constant depending on the initial values of $f(z)$ at the point $z_\theta$, where $A(z_\theta) \neq 0$ or $B(z_\theta) \neq 0$. By Definition 1, for any $\varepsilon(>0)$ and all $r \to 1^-$, we have

$$M(r,A) \leq \exp_p\{(\frac{1}{1-r})^{\sigma_{M,p}(A)+\varepsilon}\} \tag{9}$$

and

$$M(r,B) \leq \exp_p\{(\frac{1}{1-r})^{\sigma_{M,p}(B)+\varepsilon}\}. \tag{10}$$

Then, by (8)–(10) and as $\sigma_{M,p}(A) \leq \sigma_{M,p}(B)$, for the above $\varepsilon$ and all $r \to 1^-$, we have

$$
\begin{aligned}
T(r,f) \quad &\leq \quad 2\pi C[\exp_p\{(\frac{1}{1-r})^{\sigma_{M,p}(A)+\varepsilon}\} + \exp_p\{(\frac{1}{1-r})^{\sigma_{M,p}(B)+\varepsilon}\} + 1] \\
&\leq \quad \exp_{p+1}\{(\sigma_{M,p}(B) + 2\varepsilon)\log(\frac{1}{1-r})\},
\end{aligned}
$$

which implies that $\sigma_{p+1}(f) \leq \sigma_{M,p}(B) < \infty$.

Secondly, we prove $\sigma_{p+1}(f) \geq \sigma_p(B)$. Now, we rewrite (7) as

$$-B(z) = \frac{f''(z)}{f(z)} + A(z)\frac{f'(z)}{f(z)}.$$

Then, we have

$$
\begin{aligned}
T(r,B) = m(r,B) \quad &\leq \quad m(r,A) + m(r,\frac{f'}{f}) + m(r,\frac{f''}{f}) \\
&= \quad T(r,A) + m(r,\frac{f'}{f}) + m(r,\frac{f''}{f}).
\end{aligned}
\tag{11}
$$

By Lemma 1, for any $\varepsilon(>0)$, there exists a subset $E \subset [0,1)$ with $\int_E \frac{dr}{1-r} < \infty$ such that, for all $r \notin E, r \to 1^-$, we have

$$m(r,\frac{f^{(k)}}{f}) = O(\exp_{p-1}\{(\frac{1}{1-r})^{\sigma_{p+1}(f)+\varepsilon}\}) \quad (k=1,2). \tag{12}$$

If $\sigma_p(A) = \sigma_p(B)$ and $\tau_p(A) < \tau_p(B)$, then, by Definition 4, for the above $\varepsilon$ and all $r \to 1^-$, we have

$$T(r,A) \leq \exp_{p-1}\{(\tau_p(A)+\varepsilon)(\frac{1}{1-r})^{\sigma_p(A)}\} = \exp_{p-1}\{(\tau_p(A)+\varepsilon)(\frac{1}{1-r})^{\sigma_p(B)}\}, \tag{13}$$

and, by Lemma 5 and Remark 4, for sufficiently small $\varepsilon(>0)$, there exists a subset $H \subset [0,1)$ with $\int_H \frac{dr}{1-r} = \infty$ such that, for all $r \in H$, we have

$$T(r,B) > \exp_{p-1}\{(\tau_p(B) - \varepsilon)(\frac{1}{1-r})^{\sigma_p(B)}\}. \tag{14}$$

Then, by (11)–(14), for sufficiently small $\varepsilon(>0)$ and all $r \in H \backslash E, r \to 1^-$, we have

$$\exp_{p-1}\{(\tau_p(B) - \varepsilon)(\frac{1}{1-r})^{\sigma_p(B)}\}$$

$$\leq \exp_{p-1}\{(\tau_p(A) + \varepsilon)(\frac{1}{1-r})^{\sigma_p(B)}\} + O(\exp_{p-1}\{(\frac{1}{1-r})^{\sigma_{p+1}(f)+\varepsilon}\}). \tag{15}$$

Now, we may choose sufficiently small $\varepsilon(0 < 2\varepsilon < \tau_p(B) - \tau_p(A))$, and deduce, by (15), that

$$\exp_{p-1}\{(\tau_p(B) - \tau_p(A) - 2\varepsilon)(\frac{1}{1-r})^{\sigma_p(B)}\} \leq \exp_{p-1}\{(\frac{1}{1-r})^{\sigma_{p+1}(f)+2\varepsilon}\},$$

which implies that $\sigma_{p+1}(f) \geq \sigma_p(B)$. If $\sigma_p(A) < \sigma_p(B)$, then, for the above $\varepsilon$ and all $r \to 1^-$, we have

$$T(r,A) \leq \exp_{p-1}\{(\frac{1}{1-r})^{\sigma_p(A)+\varepsilon}\}, \tag{16}$$

and, by (14), for sufficiently small $\varepsilon(>0)$ and all $r \in H$, we have

$$T(r,B) \geq \exp_{p-1}\{(\frac{1}{1-r})^{\sigma_p(B)-\varepsilon}\}. \tag{17}$$

Then, by (11), (12), (16), and (17), for sufficiently small $\varepsilon(0 < 2\varepsilon < \sigma_p(B) - \sigma_p(A))$ and all $r \in H \backslash E$, $r \to 1^-$, we have

$$\exp_{p-1}\{(\frac{1}{1-r})^{\sigma_p(B)-\varepsilon}\} \leq \exp_{p-1}\{(\frac{1}{1-r})^{\sigma_p(A)+\varepsilon}\} + O(\exp_{p-1}\{(\frac{1}{1-r})^{\sigma_{p+1}(f)+\varepsilon}\}),$$

which implies $\sigma_{p+1}(f) \geq \sigma_p(B)$.

Therefore, $f(z)$ satisfies $\sigma_p(B) \leq \sigma_{p+1}(f) \leq \sigma_{M,p}(B)$. By Remark 1, we have $\sigma_{p+1}(f) = \sigma_{M,p+1}(f)$. Hence, $f(z)$ satisfies

$$\sigma_p(B) \leq \sigma_{p+1}(f) = \sigma_{M,p+1}(f) \leq \sigma_{M,p}(B).$$

Further, if $p \in \mathbf{N}_+ \backslash \{1\}$, then we have $\sigma_p(B) = \sigma_{M,p}(B)$. Consequently

$$\sigma_p(B) = \sigma_{p+1}(f) = \sigma_{M,p+1}(f) = \sigma_{M,p}(B).$$

Therefore, the proof of Lemma 6 is complete.　□

**Lemma 7.** *Let $A(z), B(z)(\not\equiv 0), F(z)(\not\equiv 0)$, and $\varphi(z)$ be analytic functions in $\boldsymbol{\Delta}$, such that $\max\{\sigma_p(A), \sigma_p(F), \sigma_p(\varphi)\} \leq \sigma_p(B)(0 < \sigma_p(B) < \infty)$, $\max\{\tau_p(S) : \sigma_p(S) = \sigma_p(B), S \in \{A, F, \varphi\}\} < \tau_p(B) < \infty$, $\sigma_{M,p}(A) \leq \sigma_{M,p}(B)$, and $\varphi(z)$ is not a solution of (1). Then, all non-trivial solutions $f(z)$ of (1) in $\boldsymbol{\Delta}$ satisfy*

$$\sigma_p(B) \leq \overline{\lambda}_{p+1}(f - \varphi) = \lambda_{p+1}(f - \varphi) = \sigma_{p+1}(f) = \sigma_{M,p+1}(f) \leq \sigma_{M,p}(B),$$

*with at most one possible exceptional solution, $f_0(z)$, satisfying*

$$\sigma_{M,p+1}(f_0) = \sigma_{p+1}(f_0) < \sigma_p(B).$$

*Further, if $p \in \mathbf{N}_+ \setminus \{1\}$, then $f(z)$ satisfies*

$$\sigma_p(B) = \overline{\lambda}_{p+1}(f - \varphi) = \lambda_{p+1}(f - \varphi) = \sigma_{p+1}(f) = \sigma_{M,p+1}(f) = \sigma_{M,p}(B),$$

*with at most one possible exceptional solution, $f_0(z)$, satisfying*

$$\sigma_{M,p+1}(f_0) = \sigma_{p+1}(f_0) < \sigma_p(B).$$

**Proof.** Firstly, we prove $\sigma_{p+1}(f) = \sigma_{M,p+1}(f) \leq \sigma_{M,p}(B)$. Let $f_1(z), f_2(z)$ be a solution base of (7). Then, we have $\sigma_{M,p+1}(f_i) \leq \sigma_{M,p}(B)(i = 1,2)$ by Lemma 6. By the elementary theory of ordinary differential equations (see, e.g., [2]), any solution of (1) can be represented as

$$f(z) = (A_1(z) + C_1)f_1(z) + (A_2(z) + C_2)f_2(z),$$

where $C_1, C_2 \in \mathbf{C}$ and $A_1(z), A_2(z)$ are analytic in $\Delta$ and are given by the system of equations

$$\begin{cases} A_1' f_1 + A_2' f_2 &= 0 \\ A_1' f_1' + A_2' f_2' &= F \end{cases}$$

satisfying $A_1' = -F f_2 W(f_1, f_2)^{-1}$ and $A_2' = F f_1 W(f_1, f_2)^{-1}$, where $W(f_1, f_2)$ is the Wronskian determinant of $f_1(z), f_2(z)$. Hence,

$$\sigma_{M,p+1}(f) \leq \max\{\sigma_{M,p+1}(F), \sigma_{M,p}(B)\}.$$

Since $\sigma_p(F) \leq \sigma_p(B) < \infty$, then $\sigma_{M,p+1}(F) = \sigma_{p+1}(F) = 0 < \sigma_p(B) \leq \sigma_{M,p}(B)$. So, $f(z)$ satisfies $\sigma_{p+1}(f) = \sigma_{M,p+1}(f) \leq \sigma_{M,p}(B)$.

Secondly, we prove $\sigma_{p+1}(f) = \sigma_{M,p+1}(f) \geq \sigma_p(B)$ with at most one possible exceptional solution, $f_0(z)$, satisfying $\sigma_{p+1}(f_0) = \sigma_{M,p+1}(f_0) < \sigma_p(B)$. On the contrary, we assume that there exist two distinct solutions $f_1(z), f_2(z)$ of (1) with $\sigma_{M,p+1}(f_i) < \sigma_p(B)(i = 1,2)$. Then, $f(z) = f_1(z) - f_2(z)$ satisfies $\sigma_{M,p+1}(f) = \sigma_{M,p+1}(f_1 - f_2) < \sigma_p(B)$. However, $f(z)$ is a non-zero solution of (7) and satisfies $\sigma_{M,p+1}(f) = \sigma_{M,p+1}(f_1 - f_2) \geq \sigma_p(B)$ by Lemma 6, which is a contradiction. Therefore,

$$\sigma_{M,p+1}(f) = \sigma_{p+1}(f) \geq \sigma_p(B)$$

with at most one possible exceptional solution, $f_0(z)$, satisfying

$$\sigma_{M,p+1}(f_0) = \sigma_{p+1}(f_0) < \sigma_p(B).$$

Therefore, we have

$$\sigma_p(B) \leq \sigma_{p+1}(f) = \sigma_{M,p+1}(f) \leq \sigma_{M,p}(B),$$

with at most one possible exceptional solution, $f_0(z)$, satisfying

$$\sigma_{M,p+1}(f_0) = \sigma_{p+1}(f_0) < \sigma_p(B).$$

Since $\varphi(z)$ is not a solution of (1)—that is, $F(z) - \varphi''(z) - A(z)\varphi'(z) - B(z)\varphi(z) \not\equiv 0$—then, by Lemma 3, we have

$$\overline{\lambda}_{p+1}(f - \varphi) = \lambda_{p+1}(f - \varphi) = \sigma_{p+1}(f) = \sigma_{M,p+1}(f).$$

Consequently, we have

$$\sigma_p(B) \leq \overline{\lambda}_{p+1}(f - \varphi) = \lambda_{p+1}(f - \varphi) = \sigma_{p+1}(f) = \sigma_{M,p+1}(f) \leq \sigma_{M,p}(B),$$

with at most one possible exceptional solution, $f_0(z)$, satisfying

$$\sigma_{M,p+1}(f_0) = \sigma_{p+1}(f_0) < \sigma_p(B).$$

Further, if $p \in \mathbf{N}_+ \backslash \{1\}$, we have

$$\sigma_p(B) = \overline{\lambda}_{p+1}(f - \varphi) = \lambda_{p+1}(f - \varphi) = \sigma_{p+1}(f) = \sigma_{M,p+1}(f) = \sigma_{M,p}(B),$$

with at most one possible exceptional solution, $f_0(z)$, satisfying

$$\sigma_{M,p+1}(f_0) = \sigma_{p+1}(f_0) < \sigma_p(B).$$

Therefore, the proof of Lemma 7 is complete.   $\square$

### 3. Proofs of Theorems 7 and 8

In this section, we denote by $E$ a subset of $[0,1)$ with $\int_E \frac{dr}{1-r} < \infty$ and by $H$ a subset of $[0,1)$ with $\int_H \frac{dr}{1-r} = \infty$, and assume that $E$ and $H$ appear not necessarily to be the same on each occasion.

**Proof of Theorem 7.** Since $\varphi(z)$ is not a solution of (1), then, by Lemma 7, all non-trivial solutions $f(z)$ of (1) satisfy

$$\sigma_p(B) \leq \overline{\lambda}_{p+1}(f - \varphi) = \lambda_{p+1}(f - \varphi) = \sigma_{p+1}(f) = \sigma_{M,p+1}(f) \leq \sigma_{M,p}(B), \tag{18}$$

with at most one possible exceptional solution, $f_0(z)$, satisfying

$$\sigma_{M,p+1}(f_0) = \sigma_{p+1}(f_0) < \sigma_p(B).$$

By (2) and (3), we have

$$\sigma_p(A_j) \leq \max\{\sigma_p(A_{j-1}), \sigma_p(B_{j-1})\} \leq \sigma_p(B) \quad (j = 1, 2, \cdots)$$

and

$$\sigma_p(B_j) \leq \max\{\sigma_p(A_{j-1}), \sigma_p(B_{j-1})\} \leq \sigma_p(B) \quad (j = 1, 2, \cdots).$$

By Lemma 1 and (2), for any $\varepsilon(> 0)$ and all $r \notin E, r \to 1^-$, we have

$$
\begin{aligned}
m(r, A_j) &\leq m(r, A_{j-1}) + O(\exp_{p-2}\{(\frac{1}{1-r})^{\sigma_p(B)+\varepsilon}\}) \\
&\leq m(r, A) + O(\exp_{p-2}\{(\frac{1}{1-r})^{\sigma_p(B)+\varepsilon}\}) \quad (j = 1, 2, \cdots).
\end{aligned}
\tag{19}
$$

On the other hand, we deduce, from (2) and (3), that

$$
\begin{aligned}
B_j &= A'_{j-1} - A_{j-1}\frac{B'_{j-1}}{B_{j-1}} + B_{j-1} \\
&= A_{j-1}(\frac{A'_{j-1}}{A_{j-1}} - \frac{B'_{j-1}}{B_{j-1}}) + A_{j-2}(\frac{A'_{j-2}}{A_{j-2}} - \frac{B'_{j-2}}{B_{j-2}}) + B_{j-2} \\
&= \sum_{k=0}^{j-1} A_k(\frac{A'_k}{A_k} - \frac{B'_k}{B_k}) + B \quad (j = 1, 2, \cdots).
\end{aligned}
\tag{20}
$$

Now, we prove that $B_j(z) \not\equiv 0$ for all $j = 1, 2, \cdots$. On the contrary, we assume that there exists some $j \in \mathbf{N}_+$ such that $B_j(z) \equiv 0$. By (19), (20), and Lemma 1, for any $\varepsilon(> 0)$ and all $r \notin E, r \to 1^-$, we have

$$
\begin{aligned}
T(r, B) = m(r, B) \quad & \leq \quad \sum_{k=0}^{j-1} m(r, A_k) + O(\exp_{p-2}\{(\frac{1}{1-r})^{\sigma_p(B)+\varepsilon}\}) \\
& \leq \quad jm(r, A) + O(\exp_{p-2}\{(\frac{1}{1-r})^{\sigma_p(B)+\varepsilon}\}) \\
& = \quad jT(r, A) + O(\exp_{p-2}\{(\frac{1}{1-r})^{\sigma_p(B)+\varepsilon}\}).
\end{aligned} \tag{21}
$$

Then by (13), (14), (16), (17), and (21), for sufficiently small $\varepsilon(> 0)$ and all $r \in H \backslash E, r \to 1^-$, we have

$$
\begin{cases}
\exp_{p-1}\{(\tau_p(B) - \tau_p(A) - 3\varepsilon)(\frac{1}{1-r})^{\sigma_p(B)}\} & \text{if } \sigma_p(A) = \sigma_p(B) \\
\leq \exp_{p-2}\{(\frac{1}{1-r})^{\sigma_p(B)+2\varepsilon}\}, & \text{and } \tau_p(A) < \tau_p(B), \\
\exp_{p-1}\{(\frac{1}{1-r})^{\sigma_p(B)-2\varepsilon}\} \leq \exp_{p-2}\{(\frac{1}{1-r})^{\sigma_p(B)+2\varepsilon}\}, & \text{if } \sigma_p(A) < \sigma_p(B),
\end{cases} \tag{22}
$$

which implies a contradiction. Hence, $B_j(z) \not\equiv 0$ for all $j = 1, 2, \cdots$.

Next, we just need to prove that $D_j(z) \not\equiv 0$ for all $j = 1, 2, \cdots$, by noting that $D_0(z) = F(z) - (\varphi''(z) + A(z)\varphi'(z) + B(z)\varphi(z)) \not\equiv 0$ since $\varphi(z)$ is not a solution of (1). On the contrary, we assume that there exists some $j \in \mathbf{N}_+$ such that $D_j(z) \equiv 0$; that is, $F_j(z) - (\varphi''(z) + A_j(z)\varphi'(z) + B_j(z)\varphi(z)) \equiv 0$. Hence, we have

$$
\begin{aligned}
F_j \quad & = \quad \varphi(\frac{\varphi''}{\varphi} + A_j\frac{\varphi'}{\varphi} + B_j) \\
& = \quad \varphi[\frac{\varphi''}{\varphi} + A_j\frac{\varphi'}{\varphi} + \sum_{k=0}^{j-1} A_k(\frac{A_k'}{A_k} - \frac{B_k'}{B_k}) + B] \quad (j = 1, 2, \cdots).
\end{aligned}
$$

By the assumption that $\varphi(z) \not\equiv 0$, we have

$$
B = \frac{F_j}{\varphi} - [\frac{\varphi''}{\varphi} + A_j\frac{\varphi'}{\varphi} + \sum_{k=0}^{j-1} A_k(\frac{A_k'}{A_k} - \frac{B_k'}{B_k})]. \tag{23}
$$

Then by (19), (23), and Lemma 1, for any $\varepsilon(> 0)$ and all $r \notin E, r \to 1^-$, we have

$$
\begin{aligned}
T(r, B) \quad & = \quad m(r, B) \leq T(r, F_j) + (j+1)T(r, A) + m(r, \frac{1}{\varphi}) + O(\exp_{p-2}\{(\frac{1}{1-r})^{\sigma_p(B)+\varepsilon}\}) \\
& \leq \quad T(r, F) + (j+1)T(r, A) + T(r, \varphi) + O(\exp_{p-2}\{(\frac{1}{1-r})^{\sigma_p(B)+\varepsilon}\}).
\end{aligned} \tag{24}
$$

By the assumptions that $\max\{\sigma_p(A), \sigma_p(F), \sigma_p(\varphi)\} \leq \sigma_p(B)$ and $\max\{\tau_p(S) : \sigma_p(S) = \sigma_p(B), S \in \{A, F, \varphi\}\} < \tau_p(B)$, similar to (13) and (16), for sufficiently small $\varepsilon(> 0)$ and all $r \to 1^-$, we have

$$
T(r, F) \leq \begin{cases}
\exp_{p-1}\{(\tau_p(F) + \varepsilon)(\frac{1}{1-r})^{\sigma_p(B)}\}, & \text{if } \sigma_p(F) = \sigma_p(B) \\
& \text{and } \tau_p(F) < \tau_p(B), \\
\exp_{p-1}\{(\frac{1}{1-r})^{\sigma_p(F)+\varepsilon}\} \leq \exp_{p-1}\{(\frac{1}{1-r})^{\sigma_p(B)-\varepsilon}\}, & \text{if } \sigma_p(F) < \sigma_p(B) \quad,
\end{cases} \tag{25}
$$

and

$$
T(r,\varphi) \leq
\begin{cases}
\exp_{p-1}\{(\tau_p(\varphi)+\varepsilon)(\dfrac{1}{1-r})^{\sigma_p(B)}\}, & \text{if } \sigma_p(\varphi)=\sigma_p(B) \\[1ex]
 & \text{and } \tau_p(\varphi)<\tau_p(B), \\[1ex]
\exp_{p-1}\{(\dfrac{1}{1-r})^{\sigma_p(\varphi)+\varepsilon}\} \leq \exp_{p-1}\{(\dfrac{1}{1-r})^{\sigma_p(B)-\varepsilon}\}, & \text{if } \sigma_p(\varphi)<\sigma_p(B)
\end{cases}
\quad (26)
$$

Denote $\tau = \max\{\tau_p(S) : \sigma_p(S) = \sigma_p(B), S \in \{A, F, \varphi\}\}$. Then, by (13), (14), (16), and (24)–(26), for sufficiently small $\varepsilon(0 < 3\varepsilon < \tau_p(B) - \tau)$ and all $r \in H\backslash E, r \to 1^-$, we have

$$
\begin{aligned}
& \exp_{p-1}\{(\tau_p(B)-\varepsilon)(\frac{1}{1-r})^{\sigma_p(B)}\} \\
\leq\ & O(\exp_{p-1}\{(\tau+\varepsilon)(\frac{1}{1-r})^{\sigma_p(B)}\}) + O(\exp_{p-1}\{(\frac{1}{1-r})^{\sigma_p(B)-\varepsilon}\}) \\
& + O(\exp_{p-2}\{(\frac{1}{1-r})^{\sigma_p(B)+\varepsilon}\}) \\
\leq\ & \exp_{p-1}\{(\tau+2\varepsilon)(\frac{1}{1-r})^{\sigma_p(B)}\},
\end{aligned}
\quad (27)
$$

which is a contradiction. Hence, $D_j(z) \not\equiv 0$ for all $j = 1, 2, \cdots$.

Then, by Theorem 4(a) and (18), we have

$$
\sigma_p(B) \leq \overline{\lambda}_{p+1}(f^{(j)} - \varphi) = \lambda_{p+1}(f^{(j)} - \varphi) = \sigma_{p+1}(f) = \sigma_{M,p+1}(f) \leq \sigma_{M,p}(B) \quad (j = 0, 1, \cdots),
$$

with at most one possible exceptional solution, $f_0(z)$, satisfying

$$
\sigma_{M,p+1}(f_0) = \sigma_{p+1}(f_0) < \sigma_p(B).
$$

Since $p \in \mathbf{N}_+\backslash\{1\}$, we have

$$
\sigma_p(B) = \overline{\lambda}_{p+1}(f^{(j)} - \varphi) = \lambda_{p+1}(f^{(j)} - \varphi) = \sigma_{p+1}(f) = \sigma_{M,p+1}(f) = \sigma_{M,p}(B) \quad (j = 0, 1, \cdots),
$$

with at most one possible exceptional solution, $f_0(z)$, satisfying

$$
\sigma_{M,p+1}(f_0) = \sigma_{p+1}(f_0) < \sigma_p(B).
$$

Therefore, the proof of Theorem 7 is complete. $\square$

**Proof of Theorem 8.** Denote $\delta = \delta(\infty, B) > 0$, then we have, by Definition 5, that for all $r \to 1^-$,

$$
T(r, B) \leq \frac{2}{\delta} m(r, B).
\quad (28)
$$

Firstly, we prove that $B_j(z) \not\equiv 0$ for all $j = 1, 2, \cdots$. On the contrary, we assume that there exists some $j \in \mathbf{N}_+$ such that $B_j(z) \equiv 0$. By (21), (28), and Lemma 1, for any $\varepsilon(> 0)$ and all $r \notin E, r \to 1^-$, we have

$$
\begin{aligned}
T(r, B) & \leq \frac{2}{\delta} m(r, B) \leq \frac{2}{\delta} \sum_{k=0}^{j-1} m(r, A_k) + O(\exp_{p-2}\{(\frac{1}{1-r})^{\sigma_p(B)+\varepsilon}\}) \\
& \leq \frac{2}{\delta} j m(r, A) + O(\exp_{p-2}\{(\frac{1}{1-r})^{\sigma_p(B)+\varepsilon}\}) \\
& \leq \frac{2}{\delta} j T(r, A) + O(\exp_{p-2}\{(\frac{1}{1-r})^{\sigma_p(B)+\varepsilon}\}).
\end{aligned}
\quad (29)
$$

Then, by (13), (14), (16), (17), and (29), for sufficiently small $\varepsilon(>0)$ and all $r \in H \backslash E, r \to 1^-$, we have (22) again, which implies a contradiction. Hence, $B_j(z) \not\equiv 0$ for all $j = 1, 2, \cdots$.

　　Secondly, we prove that $D_j(z) \not\equiv 0$ for all $j = 1, 2, \cdots$. On the contrary, we assume that there exists some $j \in \mathbf{N}_+$ such that $D_j(z) \equiv 0$. If $\varphi(z) \not\equiv 0$, then by (23), (28), and Lemma 1, for any $\varepsilon(>0)$ and all $r \notin E, r \to 1^-$, we have

$$
\begin{aligned}
T(r, B) &\le \frac{2}{\delta} m(r, B) \\
&\le \frac{2}{\delta} [m(r, F) + (j+1)m(r, A) + m(r, \frac{1}{\varphi})] + O(\exp_{p-2}\{(\frac{1}{1-r})^{\sigma_p(B)+\varepsilon}\}) \\
&\le \frac{2}{\delta} [T(r, F) + (j+1)T(r, A) + T(r, \varphi)] + O(\exp_{p-2}\{(\frac{1}{1-r})^{\sigma_p(B)+\varepsilon}\}).
\end{aligned}
\tag{30}
$$

Hence, by (13), (14), (16), (25), (26), and (30), we deduce a contradiction, similar to (27). If $\varphi(z) \equiv 0$, then, by (4) and (5), we have

$$
F'_{j-1} - F_{j-1} \frac{B'_{j-1}(z)}{B_{j-1}(z)} \equiv 0.
\tag{31}
$$

Integrating (31), we have $F_{j-1}(z) = cB_{j-1}(z)$, where $c$ is a non-zero complex constant. If $j = 1$, then $F(z) = F_0(z) = cB_0(z) = cB(z)$, which contradicts with the assumption that $\sigma_p(F) < \sigma_p(B)$ or $\sigma_p(F) = \sigma_p(B), \tau_p(F) < \tau_p(B)$. So, $j \in \mathbf{N}_+ \backslash \{1\}$, and, by (20), we have

$$
\frac{1}{c} F_{j-1} = B_{j-1} = \sum_{k=0}^{j-2} A_k (\frac{A'_k}{A_k} - \frac{B'_k}{B_k}) + B.
\tag{32}
$$

Then, by (4) and Lemma 1, for any $\varepsilon(>0)$ and all $r \notin E, r \to 1^-$, we have

$$
m(r, F_{j-1}) \le m(r, F) + O(\exp_{p-2}\{(\frac{1}{1-r})^{\sigma_p(B)+\varepsilon}\}).
\tag{33}
$$

By (28), (32), and (33), for the above $\varepsilon$ and all $r \notin E, r \to 1^-$, we have

$$
\begin{aligned}
T(r, B) &\le \frac{2}{\delta} m(r, B) \\
&\le \frac{2}{\delta} [\sum_{k=0}^{j-2} m(r, A_k) + m(r, F_{j-1})] + O(\exp_{p-2}\{(\frac{1}{1-r})^{\sigma_p(B)+\varepsilon}\}) \\
&\le \frac{2}{\delta} [(j-1)T(r, A) + T(r, F)] + O(\exp_{p-2}\{(\frac{1}{1-r})^{\sigma_p(B)+\varepsilon}\}).
\end{aligned}
\tag{34}
$$

Then by (13), (14), (16), (25), and (34), we deduce a contradiction, similar to the case for $\varphi(z) \not\equiv 0$. Hence, $D_j(z) \not\equiv 0$ for all $j = 1, 2, \cdots$. In addition, $D_0(z) = F(z) - (\varphi''(z) + A(z)\varphi'(z) + B(z)\varphi(z)) \not\equiv 0$, since $\varphi(z)$ is not a solution of (1).

　　As $B_j(z) \cdot D_j(z) \not\equiv 0$ for all $j = 0, 1, \cdots$, then, by Theorem 4(a), we obtain the result of Theorem 8. Therefore, the proof of Theorem 8 is complete.　$\square$

**Author Contributions:** All authors drafted the manuscript, and read and approved the final manuscript.

**Funding:** This project was supported by the National Natural Science Foundation of China (11761035) and the Natural Science Foundation of Jiangxi Province in China (20171BAB201002).

**Acknowledgments:** We thank the referee(s) for reading the manuscript very carefully and making a number of valuable and kind comments which improves the presentation of the manuscript.

**Conflicts of Interest:** The authors declare that they have no competing interests.

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
