# Peer review of "Value Distribution and Arbitrary-Order Derivatives of Meromorphic Solutions of Complex Linear Differential Equations in the Unit Disc"

_mathematics, doi:10.3390/math7040352_

Reviewer 1 Report

Report on the  paper: Value distribution of meromorphic solutions and their arbitrary-order derivatives of complex linear differential equations in the unit disc.

by Hai-Ying Chen and Xiu-Min Zheng

The paper is dedicated to the complex equation (1)

$$f''+A(z)f'+ B(z)f=F(z)\leqno(1)$$

where all functions are meromorphic in the open unit disc $\Delta$ of $\ \C$. Many tools are defined about  the  growth order, the type of growth, etc.  But here, due to the set of definition that is $\Delta$, $r$ is replaced by $\dsp{1\over 1-r}$.  Overall, they consider the iterated $p$-order  $\sigma_p(f)$ and the iterated $p$-exponent of convergence of the sequence of zeros $\lambda_p(f)$ and the iterated $p$-type $\tau_p(f)$.

In that way, the authors can improve certain results previously obtained by Latreuch Z. and Belaïdi B. and by Gong P. and  Xiao L. P., results which appear very complicated. Fortunately, Theorems 7 and 8 presented here are a bit easier to mind and this is the main interest of the paper.

In Theorem 7, assuming that all functions $A,\ B,\ F, \ \varphi$ are analytic in $\Delta$,  satisfying certain inequalties involving the $\sigma_p$ and $\tau_p$,then they conclude that all non-trivial sollutions of (1)   must satisfy equalities involving $\sigma_p,\ \lambda_p,...$

When all functions $A,\ B,\ F, \ \varphi$ are only meromorphic in $\Delta$, similar results are obtained, but  however are more complicated.

On the one hand, the claims obtained in Theorems 7 and 8 are a bit more simple than those they are aimed at generalizing though  they remain very specialized.

On the other hand, the proofs of both theorems are serious, rather hard, dealing with the Nevanlinna functions $T$ and $N$.

This why, finally I think the paper deserves to be published in the journal MDPI.

Author Response

Dear Prof. Jaden.lv and the referees:

Thanks a lot for your kindly supporting report on our manuscript (Manuscript ID: mathematics-455240).

Now, we have revised our manuscript and details can be seen as follows.

1.     We revise the statement of Lemma 1 such that it seems to be more explicit.

2.     We correct some typos:

(1) In Definition 4: the first three “” should be “” and the fourth  “” should be “” .

(2) In Definition 5: “respect to” should be “with respect to” .

(3) In Theorems 7 and 8: two extra spaces should be delete.

(4) In Reference 4: extra “[Cao T. B.(2009)]” should be delete.

3. We revise the format of References 8, 12, 16, 18 such that they have the uniform format with the other references.

Thanks again for your kind help!

Sincerely yours: Xiu-Min Zheng (Corresponding author)

Reviewer 2 Report

In general, the paper is well written, the material is presented in a rigorous
and concise manner and the topics are definitely of interest for the readers.

Author Response

Dear Prof. Jaden.lv and the referees:

Thanks a lot for your kindly supporting report on our manuscript (Manuscript ID: mathematics-455240).

Now, we have revised our manuscript and details can be seen as follows.

1.     We revise the statement of Lemma 1 such that it seems to be more explicit.

2.     We correct some typos:

(1) In Definition 4: the first three “” should be “” and the fourth  “” should be “” .

(2) In Definition 5: “respect to” should be “with respect to” .

(3) In Theorems 7 and 8: two extra spaces should be delete.

(4) In Reference 4: extra “[Cao T. B.(2009)]” should be delete.

3. We revise the format of References 8, 12, 16, 18 such that they have the uniform format with the other references.

Thanks again for your kind help!

Sincerely yours: Xiu-Min Zheng (Corresponding author)

This manuscript is a resubmission of an earlier submission. The following is a list of the peer review reports and author responses from that submission.